# An optimized strategy to measure protein stability highlights differences between cold and hot unfolded states

Caterina Alfano[1], Domenico Sanfelice[1], Stephen R. Martin[2], Annalisa Pastore[1,3] & Piero Andrea Temussi[1,4]

Macromolecular crowding ought to stabilize folded forms of proteins, through an excluded volume effect. This explanation has been questioned and observed effects attributed to weak interactions with other cell components. Here we show conclusively that protein stability is affected by volume exclusion and that the effect is more pronounced when the crowder's size is closer to that of the protein under study. Accurate evaluation of the volume exclusion effect is made possible by the choice of yeast frataxin, a protein that undergoes cold denaturation above zero degrees, because the unfolded form at low temperature is more expanded than the corresponding one at high temperature. To achieve optimum sensitivity to changes in stability we introduce an empirical parameter derived from the stability curve. The large effect of PEG 20 on cold denaturation can be explained by a change in water activity, according to Privalov's interpretation of cold denaturation.

[1] Department of Basic and Clinical Neurosciences, King's College London, London SE5 9RX, UK. [2] Structural Biology Science Technology Platform, The Francis Crick Institute, Mill Hill Laboratory, The Ridgeway, London NW7 1AA, UK. [3] Department of Molecular Medicine, University of Pavia, Pavia 27100, Italy. [4] Dipartimento di Scienze Chimiche, Universita' di Napoli Federico II, Napoli 80126, Italy. Correspondence and requests for materials should be addressed to P.A.T. (email: temussi@unina.it).

The stability of proteins is generally assessed in dilute solutions that, in addition to the molecule under study, contain only few other molecular species, such as those of a buffer, whose interaction with the protein is diffusion limited. However, cellular fluids contain high concentrations of macromolecules. For instance, it has been claimed that the cytoplasm of *Escherichia coli* contains several macromolecules with a global concentration from 0.20 to 0.40 g ml$^{-1}$ corresponding to up to 40% of a cell's volume[1].

The influence of crowding on protein stability has been the subject of heated controversies during the past few years, also in connection with its possible relevance for the function of proteins *in vivo*[2]. Overall, experimental evidence suggests that excluded volume effects induced by the presence of macromolecules are modest[2,3]. Some authors have drawn attention to the predominant influence of weak interactions as opposed to purely entropic effects[4]. Other authors have hypothesized but not firmly proven that the volume change associated with the transition from folded to unfolded species is much smaller than generally assumed[2]. A conclusive assessment of the effect of crowding on protein stability is hampered by various difficulties. A crucial problem comes from the uncertainty on the choice of the most suitable crowder, given the variety of crowders potentially available, and on which concentration to use. The range of crowder concentration existing in living cells is rather large and ill defined[1]; thus, it is not uncommon to find studies on crowding at concentrations lower than 10% or as large as 40%. Some studies have been performed using low molecular weight crowders whose effect can hardly be defined as crowding as it is better described as solvent perturbation[5,6]. Other authors have made use of artificial crowders such as dextran[7–9], Ficoll[10–12], polyethylene glycol (PEG)[9,13] or different proteins[14,15] that could potentially mimic the cellular environment. The role of soft interactions, evidenced by Pielak and co-workers[4,14], is probably crucial inside cells, but the use of proteins as crowders may mask the true contribution of volume exclusion, the possible relationship between the dimensions of the protein under study and those of crowders and indirect effects mediated by water activity[16]. We focused on synthetic crowders, taking extra care to ascertain whether they interact with our protein before assessing their role in volume exclusion.

The relevance of the relative dimensions of crowder and protein was only preliminarily analysed by Minton[17] in one of the early papers on crowding in the extreme case of a solute of small molecular weight in the presence of macromolecular crowders. More recently, Hoffman *et al.*[18] reconsidered the question in the case of the kinetics of protein–protein interactions but no firmly established experimental answer on the influence on protein stability exists.

Another crucial point that makes so difficult to assess the influence of crowders on protein stability is the choice of the parameter to measure stability. It is common practice to use only the midpoint of the thermal unfolding curve ($T_m$). This parameter is not very sensitive and, even worse, it can be completely misleading (*vide infra*). Here, we addressed these compelling questions by undertaking the first detailed study of the influence of crowding on the stability of yeast frataxin (Yfh1), a natural protein that undergoes cold denaturation above water freezing[10]. Szyperski *et al.*[19] have demonstrated that whenever cold and heat denaturation can be observed for the same protein, a single fit of the thermogram allows accurate determination of the heat capacity difference between the native and denatured states ($\Delta C_p$) and hence of the whole stability curve of the protein[20]. While for most proteins the cold denaturation temperature ($T_c$) is well below water freezing, Yfh1 is an ideal system for the characterization of the cold denatured species and

of the factors influencing protein thermal stability, offering the chance of using it as a tool to study protein stability in a variety of environments[21–23]. The great asset of this tool is that a substantial population of the unfolded species is at equilibrium with the folded species at all accessible temperatures[24]. In addition, the nature of the unfolded species changes with temperature, from a strongly hydrated form at lower temperatures to a weakly hydrated one at high temperature[22]. Such a situation can be compared with the setup of a sensitive electronic device, say a kind of Wheatstone Bridge[25] for protein stability, because tiny environmental variations will be detected and enhanced as a perturbation of the equilibrium.

We tested some of the most commonly used artificial crowders, that is, PEG 20, dextran 40, Ficoll 70 and Ficoll 400, to cover a wide range of molecular weights (20–400 kDa). The influence of each of these crowders was studied over a range of concentrations consistent with the composition of cellular fluids. We developed an *ad hoc* empirical approach to define a single parameter that describes thermal stability of a protein on the basis of its stability curve. We show that this parameter is helpful in describing the effect of macromolecular crowding.

Overall, our results indicate that the most effective crowders are those whose molecular weight is closest to that of the protein under study. Careful spectroscopic investigations show that all of the synthetic crowders employed have negligible interactions with Yfh1, excluding a sizeable contribution of weak nonspecific interactions. The asymmetry of the influence of crowders on high and low temperature unfolding is consistent with a more expanded nature of the low temperature unfolded state[22,26], but the unique behaviour of PEG can only be explained by a strong decrease of the water ability to solvate hydrophobic protein side chains.

## Results

**Relative size is crucial.** The crowders were chosen, among other reasons, to represent a range of different sizes, starting from one comparable to that of the protein up to much larger ones. The radius of gyration of the folded state of Yfh1 has been previously estimated at 2.1 nm, typical for a globular protein of 13.8 kDa molecular weight[23]. The dimensions of Yfh1 in the unfolded states have been measured by Aznauryan *et al.*[26] by single-molecule spectroscopy. These authors have shown that the radius of gyration of this protein, at the lowest accessible temperature, when it is a cold unfolded species, is ca. 3.3 nm. As temperature is raised from 273 K to ∼320 K, the radius of gyration of Yhf1 decreases to 2.7 nm corresponding to a continuous collapse of the unfolded chain with increasing temperature, followed by a slight reexpansion at temperatures above 320 K[26].

It is important to bear in mind that relative dimensions of the protein under study and crowders are not related to their molecular weights in a simple, linear way. It has been shown that synthetic crowders may have distinctly different volumes at different concentrations. In the case of several PEG polymers, Kozer *et al.*[27] have shown that although in the dilute regime, the polymers behave as highly solvated spheres, at higher concentrations they begin to interpenetrate one another. Thus, a synthetic crowder, such as PEG or Ficoll, mimics some properties of the medium in a eukaryotic cell. The relevant net result, in our study, is that the space occupancy of the crowder at high concentrations may be even greater than predicted by its intrinsic volume.

The influence of PEG 20, dextran 40, Ficoll 70 and Ficoll 400 at concentrations ranging from 0 up to 20% (w/v) was investigated in the temperature range 2–70 °C.

To assess the secondary structure content of Yfh1 in the presence of different crowders at varying concentrations, and thus

measure the relative populations of folded and unfolded species, we recorded thermograms, that is, thermal denaturation curves obtained from solutions of 10 μM Yfh1, by following the circular dichroism (CD) intensity at 220 nm as a function of temperature in the temperature range 2 to 70 °C and in the presence of varying concentrations of crowders (Fig. 1, upper panels). All curves show an increase of the CD signal at 220 nm as a function of increasing crowder concentration, in agreement with an increase in the population of the folded species, accompanied by a moderate increase of the high temperature melting points ($T_m$) and a more pronounced decrease of the cold denaturation temperatures ($T_c$). Altogether, these data suggest a substantial increase in the thermal stability of Yfh1 induced by increasing concentrations of the crowders, with significant differences, particularly as far as the concentration dependence is concerned.

**Volume exclusion crucially affects cold denaturation.** A deeper analysis of the influence on stability of different crowders, as yielded by the stability curves corresponding to thermal unfolding curves, is required because the population of the folded form of Yfh1 at room temperature is <100% (ref. 28).

With the assumption of a two-state equilibrium between folded and unfolded forms, the population of the folded species at any temperature, $f_{F(T)}$, is a function of $\Delta G^0(T)$, the Gibbs free energy for unfolding, that is shown as modified Gibbs–Helmholtz in equation (1).

$$\Delta G = \Delta H_m\left(1 - \frac{T}{T_m}\right) + \Delta C_p\left\{(T - T_m) - T\ln\left(\frac{T}{T_m}\right)\right\}. \quad (1)$$

Fitting of the CD thermal unfolding curves yields stability curves (Fig. 1, lower panels), from which it is possible to retrieve

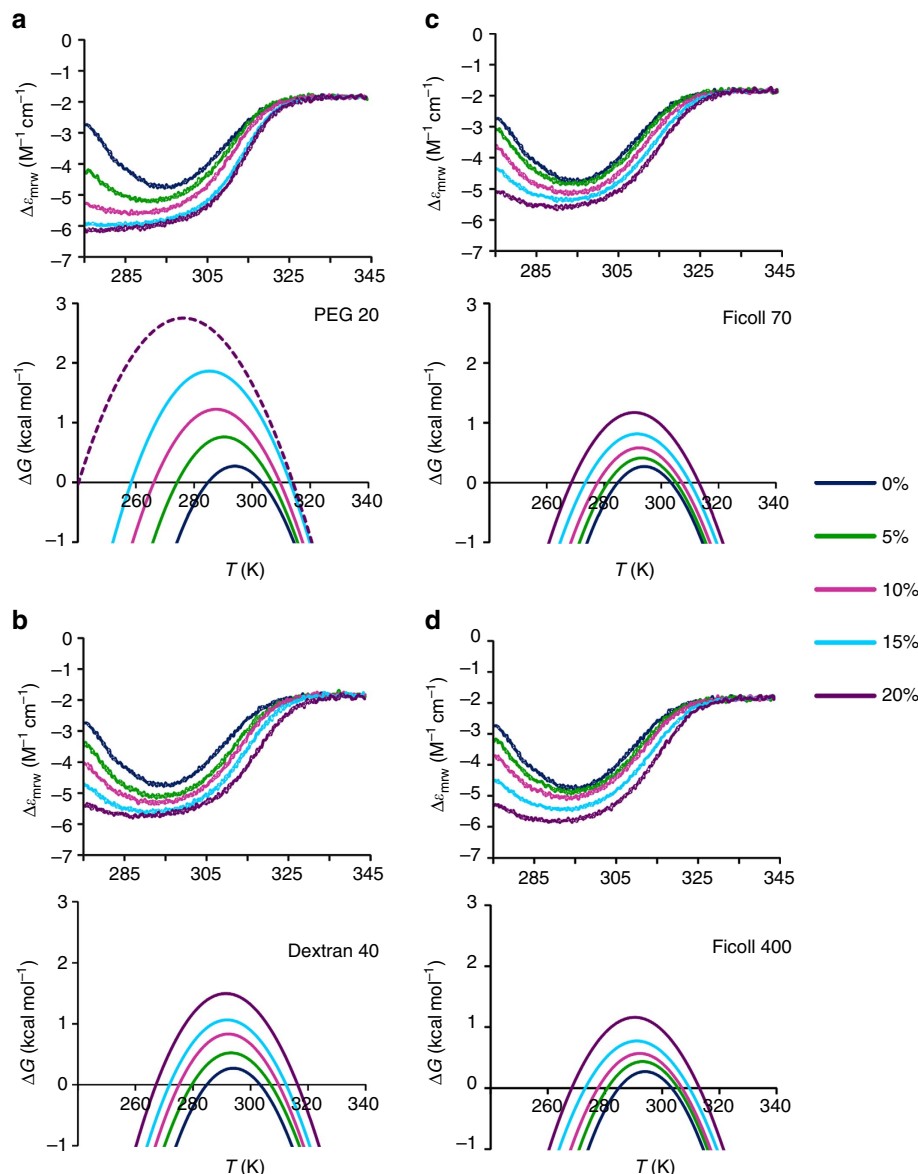

**Figure 1 | Thermal denaturation of Yfh1 in the presence of crowders.** Thermograms (upper panels) in the presence of increasing concentrations of crowders: (**a**) PEG 20, (**b**) dextran 40, (**c**) Ficoll 70 and (**d**) Ficoll 400. All curves were obtained starting from solutions of 10 μM Yfh1 in a 10 mM HEPES buffer at pH 7.0 by monitoring the CD intensity at 220 nm as a function of temperature in the temperature range 275 K to 343 °C in the presence of varying concentrations of crowders. Stability curves (lower panels) of Yfh1 corresponding to the thermograms (upper panels): (**a**) PEG 20, (**b**) dextran 40, (**c**) Ficoll 70 and (**d**) Ficoll 400. The curve corresponding to 20% w/v PEG 20 is represented as a dashed line because the fitting was extrapolated from those at lower concentrations; this is due to the observation that the thermogram (upper (**a**) panel) shows only very upward curvature at low temperatures.

thermodynamic data (Table 1). The most striking result is the difference induced by increasing amounts of crowders in the high and low temperature transitions. The maximum increase in the values of $T_m$ is similar for all crowders, 12 K for PEG 20, 13 K for dextran 40 and 10 K for both Ficolls, whereas the corresponding decrease of the values of $T_c$ ranges from ca. 44 K for PEG 20, 18 K for dextran 40 to 17 K for both Ficolls. Thus, if we assume that changes in $T_c$ and $T_m$ may be taken as *bona fide* assessments of thermal stability, for the same crowder, we must conclude that we observe a marked increase in stability at low temperature but a smaller increase in stability at high temperature.

**Protein–crowder interactions are negligible**. A possible explanation for this asymmetric effect of crowders is the presence of weak interactions of synthetic crowders with Yfh1. To minimize this risk, we had deliberately excluded proteins as crowders since they seem to be particularly prone to interactions[14,29]. However, the 'neutrality' of various synthetic crowders, after initial enthusiasm, has also been challenged to different degrees[30–33].

We thus checked neutrality with respect to soft interactions with our protein by CD and nuclear magnetic resonance (NMR). Far-ultraviolet CD spectra of Yfh1 were recorded at 25 °C in the absence and presence of 5% w/v of each crowder, that is, a concentration that should have minimal volume exclusion effect. The spectra are very similar, although with some sign of increased secondary structure content for the solutions containing either PEG 20 or dextran 40 (Supplementary Fig. 1). The increment of secondary structure content might be due to incipient volume exclusion effects rather than to soft interactions. To settle the issue we compared NMR [15]N heteronuclear single-quantum correlated (HSQC)spectra of Yfh1 in 10 mM HEPES and in 5% w/v solutions of representative crowders since NMR is more

sensitive than CD to even slight changes of the chemical environment and can thus also reveal local weak interactions (Fig. 2). As a rule of thumb, observation of chemical shift perturbations (CSPs) in NMR spectra does not necessarily indicate significant conformational changes but the absence of CSP is proof of the lack of conformational changes.

Only few peaks show minor (<0.03 p.p.m. in the proton dimension) CSP, therefore ruling out the possibility that the crowders used here significantly alter the architecture of Yfh1. The only apparent exception is that it is possible to observe, in the centre of the spectra, a decrease in the intensity of some (sharp) peaks, typical of the unfolded species of Yfh1. Their intensity is so low that the corresponding peaks of Yfh1 in the buffer solution become dominant, bursting out from the background. This may reflect a small shift of the equilibrium in favour of the folded species that can already be observed at the concentration of 5% w/v, presumably as the precocious manifestation of a volume exclusion effect (*vide infra*). The choice of showing spectra at 5% crowder concentration is based on the fact that, in doing so, we can disentangle effects arising from soft interactions and viscosity from those arising from volume exclusion effects, because it can be assumed that a 5% concentration is not sufficient to cause volume exclusion effects, whereas higher concentrations of crowders can strongly decrease the quality of NMR spectra owing to the increase of viscosity. Nonetheless, after observing larger effects for PEG 20, we extended the NMR study in the range 5 to 20% PEG 20. The spectra up to 10% concentration are of good quality, although with incipient broadening, due to the considerable increase in viscosity.

In the superposition of the spectrum in buffer with that in the presence of 10% PEG 20 (Fig. 3a) the chemical shift changes are essentially identical to those observed in the 5% comparison (Fig. 2c). The main difference is that a few peaks of the folded

**Table 1 | Relevant thermodynamic parameters of Yfh1 in the presence of crowders.**

|  | 0% | 5% | 10% | 15% | 20% |
|---|---|---|---|---|---|
|  |  | *PEG 20* |  |  |  |
| $\Delta H$ (kcal mol$^{-1}$) | 17.5 ± 0.3 | 28.0 ± 0.7 | 34.1 ± 0.5 | 41.4 ± 0.4 | 44.9 ± 0.3 |
| $\Delta C_p$ (kcal mol$^{-1}$ K$^{-1}$) | 1.85 ± 0.0 | 1.65 ± 0.03 | 1.49 ± 0.02 | 1.42 ± 0.03 | 1.12 ± 0.1 |
| $\Delta S$ (kcal mol$^{-1}$ K$^{-1}$) | 0.06 ± 0.001 | 0.09 ± 0.002 | 0.11 ± 0.002 | 0.13 ± 0.001 | 0.14 ± 0.001 |
| %f (%) | 61 | 79 | 90 | 96 | 99 |
| $T_m$ (K) | 303 ± 0.2 | 307 ± 0.3 | 310 ± 0.2 | 313 ± 0.1 | 314 ± 0.1 |
| $T_c$ (K) | 285 ± 0.3 | 274 ± 0.3 | 266 ± 0.2 | 258 ± 0.2 | 241 ± 0.2 |
|  |  | *Dextran 40* |  |  |  |
| $\Delta H$ (kcal mol$^{-1}$) | 17.5 ± 0.3 | 23.1 ± 0.8 | 29.2 ± 0.1 | 32.0 ± 0.7 | 37.4 ± 0.6 |
| $\Delta C_p$ (kcal mol$^{-1}$ K$^{-1}$) | 1.85 ± 0.01 | 1.63 ± 0.03 | 1.62 ± 0.02 | 1.510.03 | 1.44 ± 0.03 |
| $\Delta S$ (kcal mol$^{-1}$ K$^{-1}$) | 0.06 ± 0.001 | 0.07 ± 0.003 | 0.09 ± 0.001 | 0.10 ± 0.002 | 0.12 ± 0.002 |
| %f (%) | 61 | 71 | 81 | 86 | 93 |
| $T_m$ (K) | 303 ± 0.2 | 307 ± 0.4 | 310 ± 0.2 | 312 ± 0.2 | 316 ± 0.2 |
| $T_c$ (K) | 285 ± 0.3 | 280 ± 0.3 | 275 ± 0.2 | 272 ± 0.3 | 267 ± 0.2 |
|  |  | *Ficoll 70* |  |  |  |
| $\Delta H$ (kcal mol$^{-1}$) | 17.5 ± 0.3 | 20.9 ± 0.2 | 23.9 ± 0.8 | 27.1 ± 0.5 | 32.1 ± 0.6 |
| $\Delta C_p$ (kcal mol$^{-1}$ K$^{-1}$) | 1.85 ± 0.01 | 1.71 ± 0.01 | 1.57 ± 0.03 | 1.42 ± 0.02 | 1.37 ± 0.02 |
| $\Delta S$ (kcal mol$^{-1}$ K$^{-1}$) | 0.06 ± 0.001 | 0.07 ± 0.001 | 0.08 ± 0.003 | 0.09 ± 0.002 | 0.10 ± 0.002 |
| %f (%) | 61 | 67 | 73 | 80 | 88 |
| $T_m$ (K) | 303 ± 0.2 | 305 ± 0.1 | 307 ± 0.4 | 310 ± 0.2 | 313 ± 0.2 |
| $T_c$ (K) | 285 ± 0.3 | 281 ± 0.2 | 278 ± 0.3 | 273 ± 0.2 | 268 ± 0.3 |
|  |  | *Ficoll 400* |  |  |  |
| $\Delta H$ (kcal mol$^{-1}$) | 17.5 ± 0.3 | 21.3 ± 0.8 | 23.4 ± 0.7 | 25.7 ± 0.4 | 31.7 ± 0.6 |
| $\Delta C_p$ (kcal mol$^{-1}$ K$^{-1}$) | 1.85 ± 0.01 | 1.68 ± 0.02 | 1.54 ± 0.02 | 1.36 ± 0.02 | 1.35 ± 0.02 |
| $\Delta S$ (kcal mol$^{-1}$ K$^{-1}$) | 0.06 ± 0.001 | 0.07 ± 0.003 | 0.08 ± 0.002 | 0.08 ± 0.001 | 0.10 ± 0.002 |
| %f (%) | 61 | 68 | 73 | 79 | 88 |
| $T_m$ (K) | 303 ± 0.2 | 305 ± 0.4 | 307 ± 0.4 | 310 ± 0.2 | 313 ± 0.2 |
| $T_c$ (K) | 285 ± 0.3 | 281 ± 0.3 | 277 ± 0.5 | 273 ± 0.3 | 268 ± 0.2 |

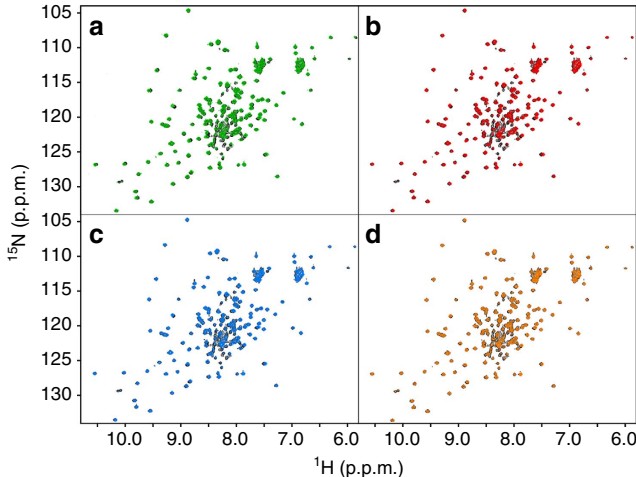

**Figure 2 | Neutrality of crowders.** Superposition of the [15]N HSQC NMR spectrum of Yfh1 at 25 °C in 10 mM HEPES buffer pH 7.0 (black) and those in the presence of 5% w/v of: (**a**) PEG 20 (green), (**b**) dextran 40 (red), (**c**) Ficoll 70 (light blue) and (**d**) Ficoll 400 (orange).

species are apparently absent. This observation is the incipient manifestation of the broadening induced by the increase in viscosity as the amount of crowder is increased; the weakest and broadest peaks are the first to be influenced. The increase in viscosity does hardly influence CD spectra but affects NMR spectra, owing to the slowing down of tumbling motions. The broadening of the peaks of the globular part of the folded species becomes extreme in the 20% spectrum, to the point that the only observable peaks are those of the intrinsically disordered N-terminus. This behaviour is very interesting since it closely parallels what we had previously observed by in-cell NMR (Fig. 3c)[34]. We can thus conclude that the disappearance of the NMR spectrum of the folded species in cell, as well as in our *in vitro* experiment, must be attributed mainly to the increase in viscosity and the ensuing reduction of tumbling, without a sizeable contribution of weak interactions. This is *per se* an important conclusion.

**Defining protein stability by $T_m$ has clear limits**. Protein stability is difficult to define using a single parameter, to the point that, in extreme cases, it may appear as an elusive concept. As pointed out by Pucci *et al.*[35] thermal stability ought to be split between 'thermodynamic stability', that is, the folding free energy $\Delta G$, and 'thermal resistance', that is, the melting temperature ($T_m$). It is common place to assume that the variation of melting temperature ($\Delta T_m$) can be used as a faithful measure of stability, implying proportionality with the change in thermodynamic stability ($\Delta\Delta G$). Becktel and Schellman[20] in their exhaustive analysis of the protein stability curve showed that when the change in free energy is small and the melting temperatures are not too close to the temperature of maximum stability ($T_S$), it is possible to relate the change in $T_m$ to the value of the perturbation free energy at the melting temperature. The relationship between changes in melting temperature and changes in free energy is linear only if we can make the assumption that the portions of the two stability curves chosen to measure the temperature increment ($\Delta T$) behave as parallel straight lines (Supplementary Fig. 2) crossing the abscissa ($T$). In actual cases this is often not true: a common cause of failure stems from the simultaneous change of the enthalpy of folding and of heat content ($\Delta C_p$). Any change in curvature ($-\Delta C_p/T$) will make the two portions not parallel. For instance, as shown by McCrary *et al.*[36] in the comparison between thermophilic Sac7d

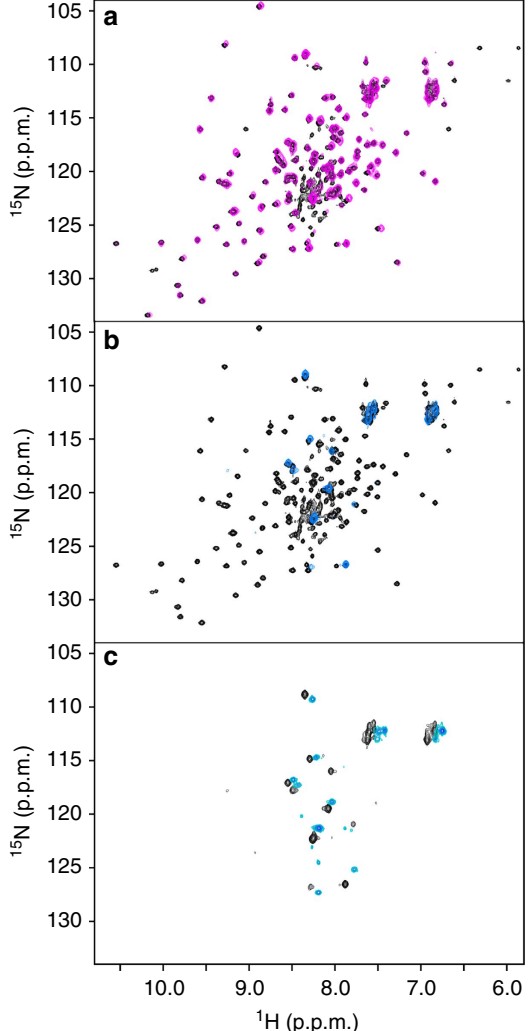

**Figure 3 | Crowders and in-cell.** Superposition of the [15]N HSQC NMR spectrum of Yfh1 at 25 °C in 10 mM HEPES buffer pH 7.0 (black) and those in the presence of (**a**) 10% w/v of PEG 20 (magenta) and (**b**) 20% w/v of PEG 20 (pale blue). (**c**) Comparison of the spectra of Yfh1 in the presence of 20% PEG 20 (black) and in *E. coli* (cyan).

and mesophilic Metmyoglobin, the two indicators yield an oxymoronic description of thermal stability: Sac7d appears to be a protein of lower thermodynamic stability, although with higher thermal resistance. It is clear, from Fig. 4a, calculated from the thermodynamic data of McCrary *et al.*[36], that Sac7d, the protein from a thermophilic organism, has a large increase in the $T_m$ with a concomitant lowering of $\Delta G_S$, the free energy difference at the temperature of maximum stability ($T_S$). It is also possible to have an increase in $\Delta G_S$, with little increase or even a decrease in $T_m$. The comparison of the stability curve of lysozyme in buffer alone and in buffer with the 5% v/v addition of propanol, calculated from the thermodynamic parameters of Velicelebi and Sturtevant[37], is shown in Fig. 4b. It is clear that the very small decrease of $T_m$ upon addition of propanol is not paralleled by a decrease in $\Delta G_S$ (thermodynamic stability). On the contrary, there is a large increase of $\Delta G_S$. These contradictory results prompted us to look for a single parameter to express stability.

**A sensitive empiric parameter to describe stability**. We propose the use of a single parameter to gauge thermal stability: the area of the stability curve between the two unfolding temperatures.

To compare areas we can take the integral of the stability curve (1) and calculate its value between $T_c$ and $T_m$.

The integral of equation (1) is given by:

$$I = -\left(\Delta H_m \frac{T^2}{2T_m}\right) + \Delta H_m T - \frac{1}{2}\Delta C_p T^2 \ln\left(\frac{T}{T_m}\right)$$
$$- \Delta C_p T_m T + \frac{3}{4}\Delta C_p T^2. \qquad (2)$$

It is easy to grasp that this parameter is an assessment of the global thermodynamic stability within the entire temperature range of protein stability, rather than at a single temperature.

In the case of the comparison between a protein from a mesophilic organism (Metmyoglobin) with a thermophilic one (Sac7d)[36] the large decrease in $\Delta H_m$ is paralleled by an increase of both $T_m$ (of 11 K) and of the integral of the stability curve ($I$), as shown in Table 2.

Thus, we have no paradox: two parameters, $T_m$ and $I$, point to greater thermal stability. In the case of the addition of propanol to the lysozyme solution the small decrease of $T_m$ is accompanied by an increase in $\Delta G_S$ of 30% but also by a much larger increase in the integral. It is fair to say that the large increase of $I$ hints at a great thermal stabilization in spite of the small decrease of $T_m$. Table 2 summarizes the values of the integral of the stability curve ($I$) between $T_c$ and $T_m$ for the examples of Fig. 4a,b, alongside the relevant thermodynamic parameters.

An interesting application of the new parameter is shown in Fig. 4c,d. Luke et al.[38] tried to correlate $T_m$ and $\Delta G_u$ values for some hyperthermostable proteins. The correlation is approximately linear, but only if values for mesophilic counterparts are included in the same graph. If one plots only values for hyperthermostable proteins the graph looks like that of Fig. 4c. It is far from linear and it is difficult to fit any curve. However, if relative areas are used instead (Fig. 4d), relative values of the integral ($I/I_o$) correlate fairly well with $T_m$ and all values fall to an exponential curve.

We have also used the new parameter to analyse our data on the stabilizing influence of alcohols[21] and found that it allows a much easier interpretation of the results (Supplementary Figs 3 and 4). Accordingly, we can use it to assess the influence of crowding on protein stability.

Using the relative increments of the area of the stability curves not only clarifies the meaning of the disparity observed between high ($T_m$) and low transition points ($T_c$) but also gives us new insights into the influence of different crowders. The comparison of the changes of $T_m$ (Fig. 5a) and of $T_c$ (Fig. 5b) illustrates why the use of this type of parameter may yield ambiguous results.

The two graphs of Fig. 5a,b offer contradictory views on the influence of crowders on the stability of Yfh1. Judging from high temperature unfolding, all crowders have apparently similar moderate influence on the stability of the protein. Conversely, judging from low temperature unfolding, it appears that all $T_c$s decrease more and those influenced by PEG 20 correspond to a very pronounced stabilizing effect on the protein. In Fig. 5c, we report the dependence of the areas of the stability curves of Yfh1 under the influence of different crowders. The areas are normalized with respect to the area of the stability curve of Yfh1 in buffer. The use of this single parameter offers an unambiguous interpretation. The dependence of the two smaller crowders is qualitatively different from those of crowders whose volume is much larger than that of Yfh1. The increase in protein stability indicated by the increase of relative areas under the influence of PEG 20 is so large as to hint at the concurrence of different causes.

## Discussion

Molecular crowding has recently come in the spotlights because of its essential role in the cell and biological fluids.

Several authors have remarked that experimental findings on the stabilization induced by crowders appear far smaller than those predicted by theoretical calculations[2,3]. This discrepancy might be attributed in part to a basic overestimation of the volume increase upon denaturation[2] but also to the relative insensitivity of melting point ($T_m$) to detect changes in stability. Depending on the combination of thermodynamic parameters, it may happen that the value of $T_m$ increases very little or even decreases, whereas thermodynamic stability increases[21,24,36]. To circumvent this difficulty we used a different single parameter, defined as the area of the stability curve to gauge protein stability that can be used in cases when cold and heat denaturation can be detected. This new parameter is very sensitive to even small changes in global protein stability. Using it we were able to determine unequivocally that volume exclusion plays an important role in the absence of weak interactions

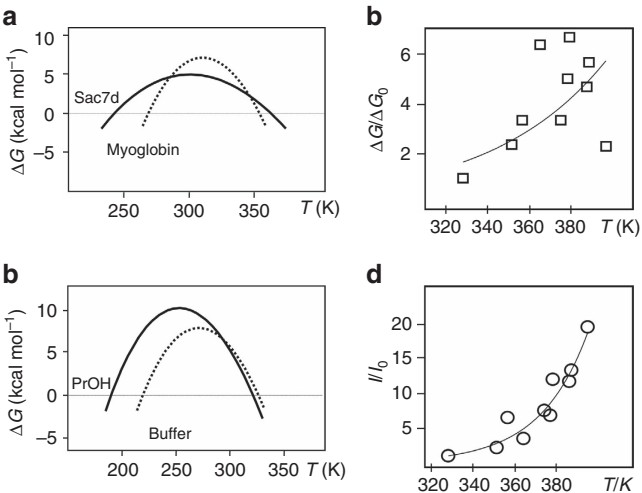

**Figure 4 | Applications of the integral parameter.** (**a**) The stability curves of Sac7d (continuous line) and Metmyoglobin (dotted line) are characterized by very different values of $\Delta C_p$. (**b**) The stability curves of lysozyme in glycine buffer at pH 2.0 (dotted line) and in the same buffer with the addition of 5% v/v propanol (PrOH, continuous line). The curves were calculated from the thermodynamic data reported in McCrary et al.[36] and in Velicelebi and Sturtevant[37] for (**a**,**b**), respectively. (**c**) Plot of relative values of $\Delta G$ (squares) for selected hyperthermophile proteins[38]. (**d**) Plot of the relative integrals (circles) for the same proteins of **c**.

**Table 2 | Thermodynamic parameters for two ambiguous cases in protein stability.**

|  | $\Delta H_m$ (kcal mol$^{-1}$) | $T_m$ (K) | $\Delta G_s$ (kcal mol$^{-1}$) | $I$(kcal K mol$^{-1}$) | Reference |
|---|---|---|---|---|---|
| Metmyoglobin | 114 | 353 | 6.97 | 384 | 36 |
| Sac7d | 58 | 364 | 5.07 | 403 | 36 |
| Lysozyme (buffer) | 91 | 325 | 7.74 | 538 | 37 |
| Lysozyme (PrOH) | 93 | 320 | 10.10 | 863 | 37 |

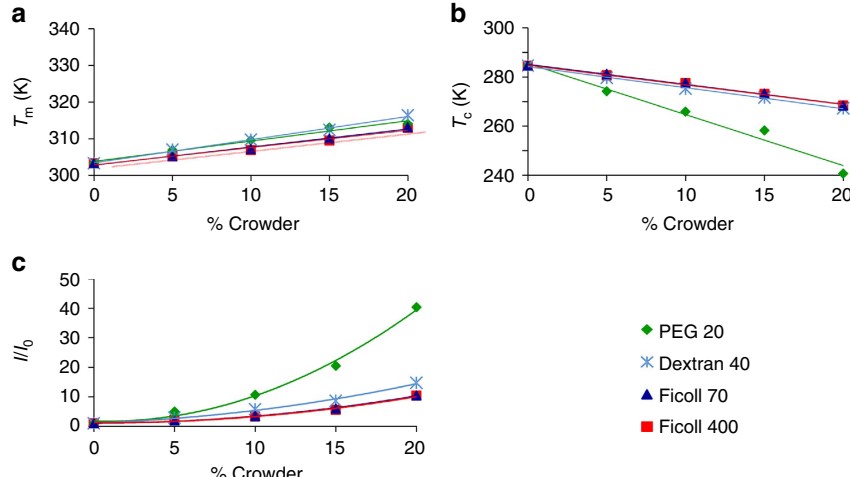

**Figure 5 | Concentration dependence.** (**a**) Variation of $T_m$ as a function of the concentration of different crowders. (**b**) Variation of $T_c$ as a function of the concentration of different crowders. (**c**) Variation of the normalized area under the stability curve of Yfh1 as a function of the concentration of different crowders.

between the crowders and the protein. We studied the influence, on the stability of Yfh1, of different macromolecular crowders at concentrations ranging between 5 and 20% w/v. This comparison highlights the importance of the relative dimensions of the protein under study and of the crowders. Crowders whose molecular weight is closer to that of the protein under study appear to be more effective than larger crowders.

A very important result emerging from our study is that the influence of crowders on high and low temperature unfolding is dramatically different. The possibility of observing cold denaturation reveals that low temperature unfolding is much more sensitive to the presence of macromolecules that affect volume availability. The most likely interpretation of this marked asymmetry is that the volume of the low temperature unfolded species is larger than that at high temperature. This interpretation is consistent with our previous observations on the difference between the two unfolded states[22,23], mainly based on a strong hydration of the low temperature species that leads to a more expanded state. This finding was later confirmed by a direct FRET measurement[26].

The influence of PEG 20 on cold denaturation of Yfh1 is so larger than those of the other crowders that it cannot be attributed solely to the larger volume of the low temperature unfolded species. Having ruled out a direct interaction of PEG 20 with our protein, the most likely cause of the marked asymmetry observed between low and high temperature behaviour of Yfh1 in the presence of PEG must be sought in the indirect effect on water activity. The ability of PEG polymers to lower water activity is routinely exploited in protein crystallization[39,40]. Its asymmetric effect on cold and hot denaturation is consistent with the mechanism of cold denaturation based on preferential hydration of the hydrophobic groups of the protein core[41].

It is fair to ask whether the behaviour of Yfh1 is idiosyncratic. How general are our findings? Is the dependence of stability on crowder's relative size a unique property of this protein or a general property of many if not all proteins? Can we extrapolate it to other proteins? The answer is yes, with the proviso that it is generally difficult to observe the influence on low temperature unfolded species directly, simply because cold denaturation transition occurs, for most proteins, at too low a temperature to allow direct observation. However, although it is not generally possible to observe the low temperature transition, it is often possible to measure $\Delta C_p$ accurately from calorimetric

measurements. Thus, we can take advantage of the full stability curve and of the related integral parameter. It remains to be seen whether the low temperature unfolded species is more expanded than the high temperature one in all proteins. It is quite possible that the volume change upon unfolding, both at high and low temperature, can vary widely among different proteins[2].

It is important to emphasize that the analysis of the influence of crowders on Yfh1 was greatly facilitated by gauging stability with the aid of the integral of the stability curve. We can thus conclude that our study sets a clear answer on the question of the influence of volume exclusion, also in relation with the optimal size of a crowder and demonstrates that the use of integrals of the stability curve is a very useful addition to the parameters commonly employed to describe the thermal stability of proteins.

## Methods

**Protein preparation.** Yfh1 was expressed in *E. coli* and purified as described by He *et al.*[42] Purity of the recombinant proteins was checked by SDS–polyacrylamide gel electrophoresis after each step of the purification.

All crowding agents, PEG 20, dextran 40, Ficoll 70 and Ficoll 400 were purchased from Sigma Aldrich. Concentrated solutions to be used as mother solutions to prepare samples at different crowder concentrations were examined for their potential influence on the final ionic strength and pH.

**In-cell experiments.** Yfh1, cloned in pET21a vectors, was transformed in BL21(DE3) *E. coli* cells and selected for transformation in ampicillin plates, according to previously published protocols[42]. As previously described[34], single colonies were grown overnight at 37 °C in 3 ml of Luria Broth medium and 2 ml of the cultures were used to inoculate 100 ml of fresh medium. When the cells reached the desired optical density, they were collected by centrifugation at 3,600 *g* for 15 min at room temperature. The pellet was then resuspended in 100 ml of [15]N enriched minimal medium, incubated in a rotary shaker at 37 °C for 10 min and induced with isopropylthiogalactoside (0.5 mM).

The final pellet was resuspended in 500 ml of M9 medium mixed with 50 ml of $D_2O$, before transferring it into standard 5 mm NMR tubes using a Pasteur pipette and used immediately for NMR experiments.

**NMR spectroscopy.** NMR spectra were recorded on a Bruker AVANCE operating at 600 MHz 1H frequency. Measurements were carried out using a 15 N uniformly labelled protein at a concentration of 70 μM in a 20 mM HEPES buffer at pH 7.0 and in the same buffer supplemented with 5% of PEG 20, dextran 40, Ficoll 70 or Ficoll 400. Water suppression was achieved by the WATERGATE pulse-sequence[43], HSQC spectrum experiments were used as described by Bax *et al.*[44] The spectra were processed using the NMRPipe program[45]. Baseline correction was applied when necessary.

**Far-ultraviolet CD measurements.** Samples were prepared using a Yfh1 concentration of 10 μM in a 10 mM HEPES buffer at pH 7.0 and with varying

concentrations of PEG 20, dextran 40, Ficoll 70 or Ficoll 400. Baseline correction was obtained by subtraction of the appropriate buffer spectrum. Thermal unfolding curves were obtained by monitoring the ellipticity at 220 nm using a Jasco J-815 CD spectropolarimeter equipped with a Jasco CDF-4265/15 Peltier unit. The 1 mm path length cells (Hellma) were used and a heating rate of 2 °C min$^{-1}$ in the temperature range 2–70 °C.

**Equations to describe stability.** To have an accurate insight into protein stability, it is desirable to be able to measure its whole stability curve. This curve is the plot of $\Delta G$, the difference of free energy between the folded and unfolded species, as calculated from a modified Gibbs–Helmholtz equation (1) in which the reference temperature is the midpoint of the high temperature transition ($T_m$), as a function of temperature. For most proteins, fitting thermograms based on spectroscopic measurements, for example, CD and NMR spectroscopies, yields accurate vales for $T_m$ and $\Delta H_m$. However, these fits are totally insensitive to the value of $\Delta C_p$, unless it is possible to observe two unfolding transitions, that is, cold denaturation and heat denaturation. The occurrence of cold denaturation is obvious from the stability curve of any protein[20]. The bell-shaped form of the curve implies that destabilization of the folded species happens at the two zero points of $\Delta G$, as the temperature varies from room temperature in either direction, that is, to higher or lower temperatures. All thermodynamic parameters can then be extracted from a plot of $\Delta G$ against temperature, by means of an original fitting program[46]. The calculation of errors on the parameters is based on the fitting procedure. The errors in $\Delta C_p$ and other parameters are modest because the temperature dependence of $\Delta G$ is quite sensitive to the value of $\Delta C_p$ (proportional to the curvature of the stability curve).

**Data availability.** The data that support the findings of this study are available from the corresponding author on reasonable request.

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

## Acknowledgements

The work was supported by MRC (U117584256).

## Author contributions

All of the authors contributed to the research design and data analyses. C.A. performed most of the experiments described here. D.S. and C.A. performed the data fitting. S.R.M. and C.A. performed the extraction of the thermodynamic data. P.A.T. and A.P. wrote the manuscript.

## Additional information

**Competing interests:** The authors declare no competing financial interests.

**Publisher's note**: 

