## [Peer review file · Nature Communications]

Reviewers' comments:

Reviewer #1 (Remarks to the Author):

Alfano et al. report an interesting study of macromolecular crowding effects on protein stability. Using their well-studied model system – frataxin – the authors provide convincing evidence that the integral of the stability curve is a better gauge of stability than the more commonly used melting point (T_m). This parameter, however, is accessible only for proteins that undergo both cold and heat denaturation.

Using CD spectroscopy, it was shown that four different crowders induce significant increases in protein stability. Protein-crowder interactions were assessed by NMR. This aspect of the paper requires further work. The crowders were assessed by NMR at 5 % only, while up to 20 % was used in the CD experiments. Presumably, the protein-crowder interactions could become more important at the higher concentrations of crowder? There are chemical shift differences even at 5 % PEG, suggesting some binding. The authors should extend the NMR study to include a range of PEG concentrations up to 20 %. Also, figure 2 can be improved by showing the reference spectrum in black. The spectra in panel C are indistinguishable.

Considering that one of the main results concerns the relative size of test protein and crowder additional details are required. What is the molecular weight of frataxin? What are the hydrodynamic radii of the folded and unfolded states? How do the protein sizes compare with the synthetic polymers? This analysis should consider that the polymer “size” varies as a function of concentration (e.g. Kozer et al. Biophys J 2007). A table of these parameters could be useful.

Other points:

The title is long and unwieldy.

“our protein” replace with “the test protein”

“the crowder volume” replace with “the crowder’s size”. Use of ‘volume’ is potentially confusing in the context of ‘volume exclusion’.

Last sentence of abstract is vague, “a far greater sensitivity” of what?

Results, first paragraph “There was no evidence of significant CD spectral changes at 25 °C.” This sentence seems to be out of place.

Reviewer #2 (Remarks to the Author):

On reading with attention the manuscript by Temussi and colleagues, it is not clear what is the “new take home message” of the investigation. It has been shown, by the same authors, by about ten years, that yeast frataxin, Yfh1 undergoes cold denaturation at temperatures well above 0 °C [JACS 129 (2007) 5374, ref 28 of the manuscript], and it has been shown that macromolecular crowders, specifically Ficoll 70, are able to increase the stability of Yfh1 [Phys.Biol. 10 (2013) 045002, ref 10 of the manuscript]. The novelty of the present study would be that different crowders have been considered, PEG 20, Dextran 40, Ficoll 70 and Ficoll 400, but their effect is qualitatively the same.

In addition, there are other weak-points.

1. Temussi and colleagues show that these crowders do not interact with Yfh1 by performing HSQC NMR measurements in solutions of 5% w/v (see Figure 2 of the manuscript), and so they claim that their study is able to estimate solely the excluded volume contribution to protein stability afforded by macromolecular crowders. This claim cannot be considered right because such crowders afford a significant stabilization of Yfh1 at higher concentrations, 15 and 20% w/v, and, for such conditions, there are no experimental data ruling out the possibility of interactions between Yfh1 and crowders.

2. There are sentences indicating an error in understanding the well-established volume behaviour of globular proteins. For instance, at the end of page 2 of the manuscript, it is written: “Other authors have hypothesized but not firmly proven that the volume change associated with the transition from folded to unfolded species is much smaller than generally assumed [the reference is at an article by Politou and Temussi in *Curr.Opin.Struct.Biol.* 30 (2015) 1].” Actually, it is well-known that the volume change upon unfolding is very-very small and negative: the unfolded state has a partial molar volume smaller than the folded state; this is the well-known protein volume paradox [see, for instance, the reviews by Royer in *BBA* 1595 (2002) 201, and Chalikian in *Annu.Rev.Biophys.Biomol.Struct.* 32 (2003) 207]. This is why very high hydrostatic pressures are able to destroy the folded state of globular proteins. The problem is that, in trying to explain the effect of macromolecular crowders, usually the two states of globular proteins are modelled as two spheres with different gyration radius, and the latter is assumed to be larger for the sphere representing the unfolded state. This procedure, even though commonly and widely used, is not correct and indeed provides stabilization effects larger than those experimentally measured.

3. Temussi and colleagues propose and claim that the area of the stability curve, the parabola-like curve of the unfolding Gibbs energy versus temperature, should be a measure of the protein stability better than the single value of the unfolding Gibbs energy at the temperature of maximal stability (i.e., the maximum of the parabola-like curve). This proposal is not entirely new, since Temussi advanced it in *Biophys.Chem.* 208 (2016) 4, and does not seem to provide any actual advantage. The comparisons shown in Figure 3 of the manuscript cannot be considered a demonstration. In particular, Figure 3 of the article by Luke et al., *FEBS J.* 274 (2007) 4023, ref 39 of the manuscript, indicates the occurrence of a clear correlation between the values of the unfolding Gibbs energy at the temperature of maximal stability and the high unfolding temperature for several globular proteins. On this matter, see also the article by Rees and Robertson in *Protein Sci.* 10 (2001) 1187.

Therefore, there is no clear advantage in using the “area” of the stability curve. In addition, usually, stabilization refers to an increase of the high unfolding temperature, not to a decrease of the low unfolding temperature.

4. Temussi and colleagues do not propose any explanation of why low temperature unfolding is more affected by crowders with respect to high temperature unfolding. They have written the following sentence: “The asymmetry of the influence of crowders on high and low temperature unfolding is consistent with a more expanded nature of the low temperature unfolded state.” This could be considered an explanation if supported by calculations of a reliable model for the effect of macromolecular crowders.

On these grounds I do not support publication of the manuscript in Nature Communications.

Reviewer #3 (Remarks to the Author):

The manuscript „The many facets of the influence of macromolecular crowders on protein stability reveal that low temperature unfolding is intrinsically different from high temperature unfolding“ by Alfano et al. reports on crowding studies on the folding stability of yeast frataxin. The innovative aspect of this study is that here crowding effects on both cold and heat denaturation are considered. The authors rationalize their experimental findings by analyzing the crowding effect on the temperature dependence of the folding free energy. In particular, they choose to quantify the free energy changes by a single parameter, which they define as the area enclosed by the individual curves with respect to dilute solution. They show that this is a much more powerful approach compared to numerous previous studies (summarized by their recent review in *Curr. Opin. Struct. Biol.*) that considered the effects of crowding agents on the melting temperature only, which could be misleading. Thus, this is a very insightful study that will be interesting to a broad community of scientist that study in general the effect of cosolutes on protein folding. Thus, I highly recommend its publication but have three suggestions that should be addressed before publication.

- 1.) The authors have great expertise in analyzing the folding of yeast frataxin from their previous work and the data appear to be robust and convincing. However, in the manuscript the authors do not present any error discussion e.g. of the thermodynamic quantities in Table 1. This would be very useful to compare to other approaches to determine (and extrapolate) $dG(T)$ e.g. by NMR or calorimetry (which the authors suggest in the discussion to be used for analysis of proteins other than yeast frataxin). Especially for the high concentrations of crowders in Figure 1 it seems difficult to me to determine T_c .
- 2.) The authors exclude from their NMR measurements non-specific interactions between the crowder and the protein. However, they measure significant changes in enthalpy (Table 1) upon increase of crowder concentration. Please discuss this.
- 3.) It may be beneficial to compare their approach to quantify the crowder induced changes of $dG(T)$ by the integral I/I_0 to a recently published similar approach by Senske et al., *Phys. Chem. Chem. Phys.*, 18, 29698-29708, 2016. In their work the authors used 2 parameters to quantify the respective shifts of the curves for different cosolutes including different salts. This allows to distinguish a shift to the “right” as for e.g. sorbitol from a shift to the left e.g. propanol which is not possible for the integral approach. Maybe the authors can extend their approach to such cases.

Reviewer #1 (Remarks to the Author) & answers

Alfano et al. report an interesting study of macromolecular crowding effects on protein stability. Using their well-studied model system 13; frataxin 13; the authors provide convincing evidence that the integral of the stability curve is a better gauge of stability than the more commonly used melting point (T_m). This parameter, however, is accessible only for proteins that undergo both cold and heat denaturation.

Using CD spectroscopy, it was shown that four different crowders induce significant increases in protein stability. Protein-crowder interactions were assessed by NMR. This aspect of the paper requires further work. The crowders were assessed by NMR at 5 % only, while up to 20 % was used in the CD experiments. Presumably, the protein-crowder interactions could become more important at the higher concentrations of crowder? There are chemical shift differences even at 5 % PEG, suggesting some binding. The authors should extend the NMR study to include a range of PEG concentrations up to 20 %. Also, figure 2 can be improved by showing the reference spectrum in black. The spectra in panel C are indistinguishable.

We thank the reviewer for the kind words of appreciation of our work.

We agree that showing the NMR spectrum in the presence of a range of PEG 20 concentrations may be helpful. The reason why we showed only spectra at 5% concentration in the original version of the manuscript was that in doing so, we wanted to disentangle effects arising from soft interactions and viscosity from those (presumably) arising from volume exclusion effects: it can be assumed that a 5% concentration is not sufficient to cause volume exclusion effects.

We have now extended the NMR study in the range 5% to 20% PEG 20. The spectra up to 10 % concentration are of good quality, albeit with incipient broadening, due to the considerable increase in viscosity which is expected.

The comparison between the HSQC NMR spectra of Yfh1 in buffer and in the presence of 10% PEG 20 shows that the protein is completely folded, yet the chemical shift changes are modest (new Figure 3). At 20% concentration, the spectrum of the protein disappears almost totally except for the N-terminal unstructured part. These results are FANTASTIC!!! The spectrum at 20% PEG matches perfectly what we observed for the spectrum of Yfh1 in cell (Popovic, M., Sanfelice, D., Pastore, C., Prischi, F., Temussi, P. A. & Pastore, A. Selective observation of the disordered import signal of a globular protein by in-cell NMR: the example of frataxins. *Protein Sci.* 24, 996-1003 (2015)). This clarifies a completely new aspect: the disappearance of the NMR spectrum of the folded species in cell, as well as in our in vitro experiment, can be attributed mainly to the increase in viscosity and the ensuing reduction of tumbling, without a sizeable contribution of weak interactions.

We have included this information in the revised version of the manuscript:

Only few peaks show minor (<0.03 ppm in the proton dimension) chemical shift perturbation, therefore ruling out the possibility that the crowders used here significantly alter the architecture of Yfh1. The only apparent exception is that it is possible to observe, in the center of the spectra, a decrease in the intensity of some (sharp) peaks, typical of the unfolded species of Yfh1. Their intensity is so low that the corresponding peaks of Yfh1 in the buffer solution become dominant, bursting out from the background. This may reflect a small shift of the equilibrium in favor of the folded species, which can already be observed at the concentration of 5% w/v, presumably as the precocious manifestation of a volume exclusion effect (*vide infra*). The choice of showing spectra at 5% crowders concentration is based on the fact that, in doing so, we can disentangle effects arising from soft interactions and viscosity from those arising from volume exclusion effects, because it can be assumed that a 5% concentration is not sufficient to cause volume exclusion effects, whereas higher concentrations of crowders can strongly decrease the quality of NMR spectra owing to the

increase of viscosity. Nonetheless, after observing larger effects for PEG 20, we extended the NMR study in the range 5% to 20% PEG 20. The spectra up to 10 % concentration are of good quality, albeit with incipient broadening, due to the considerable increase in viscosity. In the superposition of the spectrum in buffer with that in the presence of 10% PEG 20 (Figure 3a) the chemical shift changes are essentially identical to those observed in the 5% comparison (Figure 2c). The main difference is that a few peaks of the folded species are apparently absent. This observation is the incipient manifestation of the broadening induced by the increase in viscosity as the amount of crowder is increased; the weakest and broadest peaks are the first to be influenced. The increase in viscosity does hardly influence CD spectra but affects NMR spectra, owing to the slowing down of tumbling motions. The broadening of the peaks of the globular part of the folded species becomes extreme in the 20% spectrum, to the point that the only observable peaks are those of the intrinsically disordered N-terminus. This behaviour is very interesting since it closely parallels what we had previously observed by in-cell NMR (Figure 3 c)³⁵. We can thus conclude that the disappearance of the NMR spectrum of the folded species in cell, as well as in our in vitro experiment, must be attributed mainly to the increase in viscosity and the ensuing reduction of tumbling, without a sizeable contribution of weak interactions. This is *per se* an important conclusion.

It is also important to draw attention to the fact that the chemical shift perturbation observed is so small that, in most studies on protein-protein interaction, it would be dismissed or attributed to a generic solvent effect.

Figure 2 has been modified as suggested by the reviewer.

Considering that one of the main results concerns the relative size of test protein and crowder additional details are required. What is the molecular weight of frataxin? What are the hydrodynamic radii of the folded and unfolded states? How do the protein sizes compare with the synthetic polymers? This analysis should consider that the polymer "size " varies as a function of concentration (e.g. Kozier et al. Biophys J 2007). A table of these parameters could be useful.

We agree; and have added this important information, not as a table but as an introduction to Results:

< The crowders were chosen, among other reasons, to represent a range of different sizes, starting from one comparable to that of the protein up to much larger ones. The radius of gyration (Rg) of the folded state of Yfh1 has been previously estimated at 2.1 nm, typical for a globular protein of 13.8 kDa MW (Adrover et al., 2012). The dimensions of Yfh1 in the unfolded states have been measured by Aznauryan et al. (2013) by single molecule spectroscopy. These authors have shown that the radius of gyration of this protein, at the lowest accessible temperature, when it is a cold unfolded species, is ca. 3.3 nm. As temperature is raised from 273 to ~320 K the Rg of Yfh1 decreases to 2.7 nm corresponding to a continuous collapse of the unfolded chain with increasing temperature, followed by a slight re-expansion at temperatures above 320 K

It is important to bear in mind that relative dimensions of the protein under study and crowders are not related to their molecular weights in a simple, linear way. It has been shown that synthetic crowders may have distinctly different volumes at different concentrations. In the case of several PEG polymers, Kozier et al. (2006) have shown that although in the dilute regime, the polymers behave as highly solvated spheres, at higher concentrations they begin to interpenetrate one another. Thus, a synthetic crowder, such as PEG or Ficoll, mimics some properties of the medium in a

eukaryotic cell. The relevant net result, in our study, is that the space occupancy of the crowder at high concentrations may be even greater than predicted by its intrinsic volume.>

Ref #1 says

Other points:

The title is long and unwieldy.

The reviewer is absolutely right. We have modified the title; "A new strategy to measure protein stability highlights the difference between cold and hot unfolded states of a protein"

"our protein " replace with "the test protein "

Done

"the crowder volume" replace with "the crowder 's size ". Use of "volume "is potentially confusing in the context of "volume exclusion ".

Done

Last sentence of abstract is vague, "a far greater sensitivity" of what?

The reviewer is right. We have modified the phrase as follows: "The use of a new single empirical parameter derived from the stability curve allows a far greater sensitivity to changes in stability induced by crowders with respect to unfolding temperature."

Results, first paragraph "There was no evidence of significant CD spectral changes at 25 °C. " This sentence seems to be out of place.

We have omitted the sentence.

Reviewer #2 (Remarks to the Author) & answers

On reading with attention the manuscript by Temussi and colleagues, it is not clear what is the "new take home message " of the investigation. It has been shown, by the same authors, by about ten years, that yeast frataxin, Yfh1 undergoes cold denaturation at temperatures well above 0 °C [JACS 129 (2007) 5374, ref 28 of the manuscript].

and it has been shown that macromolecular crowders, specifically Ficoll 70, are able to increase the stability of Yfh1 [Phys.Biol. 10 (2013) 045002, ref 10 of the manuscript]

We believe that the reviewer might have missed the main message of this work. It is true that, ten years ago, we found the first protein undergoing cold denaturation under (quasi) physiological conditions above zero degrees. Here, we are NOT proposing this finding again. We are using this protein as a tool, as done several times in the last ten years to gain new insights into protein stability and properties. The novelty is that we are examining quantitatively the influence of crowding on both transitions (low and high temperature).

The data referred to in [Phys.Biol. 10 (2013) 045002, ref 10 of the manuscript] were preliminary and inconclusive data of a small local symposium. They presented the effect of only one crowder (Ficoll 70) at three concentrations. As customary with proceedings of meetings, they are followed by a full study.

It is possible that this referee took part in the symposium and was impressed by these data.

The novelty of the present study would be that different crowders have been considered, PEG 20, Dextran 40, Ficoll 70 and Ficoll 400, but their effect is qualitatively the same

It is NOT true that the effects of PEG 20, Dextran 40, Ficoll 70 and Ficoll 400 are qualitatively the same. The effects of PEG 20 and Dextran 40 are drastically different and this could be shown only by using the new area parameter.

1. Temussi and colleagues show that these crowders do not interact with Yfh1 by performing HSQC NMR measurements in solutions of 5% w/v (see Figure 2 of the manuscript), and so they claim that their study is able to estimate solely the excluded volume contribution to protein stability afforded by macromolecular crowders. This claim cannot be considered right because such crowders afford a significant stabilization of Yfh1 at higher concentrations, 15 and 20% w/v, and, for such conditions, there are no experimental data ruling out the possibility of interactions between Yfh1 and crowders.

This point has been raised also by referee #1 and is quite reasonable; we now show the spectra of Yfh1 at 10% and 20% PEG 20 (see answer to ref #1).

However, we had chosen to show the 5% spectra as the most significant because at 5% it is fair to assume that there is not yet an excluded volume effect; so, we are seeing the influence of “pure” interactions on the NMR spectra.

2. There are sentences indicating an error in understanding the well-established volume behaviour of globular proteins. For instance, at the end of page 2 of the manuscript, it is written: "Other authors have hypothesized but not firmly proven that the volume change associated with the transition from folded to unfolded species is much smaller than generally assumed [the reference is at an article by Politou and Temussi in *Curr.Opin.Struct.Biol.* 30 (2015) 1]. " Actually, it is well-known that the volume change upon unfolding is very-very small and negative: the unfolded state has a partial molar volume smaller than the folded state; this is the well-known protein volume paradox [see, for instance, the reviews by Royer in *BBA* 1595 (2002) 201, and Chalikian in *Annu.Rev.Biophys.Biomol.Struct.* 32 (2003) 207]. This is why very high hydrostatic pressures are able to destroy the folded state of globular proteins. The problem is that, in trying to explain the effect of macromolecular crowders, usually the two states of globular proteins are modelled as two spheres with different gyration radius, and the latter is assumed to be larger for the sphere representing the unfolded state. This procedure, even though commonly and widely used, is not correct and indeed provides stabilization effects larger than those experimentally measured.

Outside the “crowding community” the ideas of Chalikian are well known. We were the first to draw the attention of the “crowding community” to this fact [Politou and Temussi in *Curr.Opin.Struct.Biol.* 30 (2015)]. In the present manuscript, we used deliberately a low profile not to sound too polemic with this community.

In addition, we DID show that IF the volume of the unfolded species IS larger, as is the case of the cold species of our protein, the volume exclusion effect can be observed clearly.

3. Temussi and colleagues propose and claim that the area of the stability curve, the parabola-like curve of the unfolding Gibbs energy versus temperature, should be a measure of the protein stability better than the single value of the unfolding Gibbs energy at the temperature of maximal stability (i.e., the maximum of the parabola-like curve). This proposal is not entirely new, since Temussi advanced it in *Biophys.Chem.* 208 (2016) 4, and does not seem to provide any actual advantage. The comparisons shown in Figure 3 of the manuscript cannot be considered a demonstration. In particular, Figure 3 of the article by Luke et al., *FEBS J.* 274 (2007) 4023, ref 39 of the manuscript, indicates the occurrence of a clear correlation between the values of the unfolding Gibbs energy at the temperature of

maximal stability and the high unfolding temperature for several globular proteins. On this matter, see also the article by Rees and Robertson in *Protein Sci.* 10 (2001) 1187. Therefore, there is no clear advantage in using the "area" of the stability curve. In addition, usually, stabilization refers to an increase of the high unfolding temperature, not to a decrease of the low unfolding temperature.

The proposal advanced in *Biophys.Chem.* 208 (2016) 4 was purely qualitative, with no calculation of the area parameter proposed in the manuscript.

As for the "no clear advantage" in the use of this parameter, the referee apparently refutes the reality principle: the points referring to thermophiles in the line in Figure 3 of the article by Luke et al., *FEBS J.* 274 (2007) 4023 are completely misaligned; the line is based solely on the mesophile points.

This is the original figure of the quoted reference:

Fig. 3. T_M versus ΔG_U values for hyperthermostable proteins in Table 2 (filled circles, those for which both values are known; cpn10 proteins excluded) along with their mesophilic counterparts (open circles, data mentioned in the text). The plot shows that the two parameters are correlated (solid line) for both sets of proteins.

The filled circles (inside the blue ellipse), if taken alone, do not fall on a line, certainly not on the same line as mesophiles (open circles, inside the red ellipse). Yet, the referee sees them aligned.

Even refusing to use a decrease of the low unfolding temperature as an indication of stability is a way to abandon the reality principle. It is enough to look at the stability curve. The increase of the high unfolding temperature is simply an indication of increased THERMAL RESISTANCE. It is usually taken as a measure of thermal stability on the *assumption* that it is proportional to an increase of free energy, but Bechtel & Schellman have demonstrated that this is true only under certain assumptions (see S.I.).

4. Temussi and colleagues do not propose any explanation of why low temperature unfolding is more affected by crowders with respect to high temperature unfolding. They have written the following sentence: "The asymmetry of the influence of crowders on high and low temperature unfolding is consistent with a more expanded nature of the low temperature unfolded state." This could be considered an explanation if supported by calculations of a reliable model for the effect of macromolecular crowders.

We were based on strong experimental evidence provided both by our own group and by other measurements of the radius of gyration of Yfh1 at low and high temperatures (Adrover et al. XXX and Schuler). These results find in the theory proposed by Allen

Minton long ago a fair common ground for interpretation. Our purpose was in fact to validate the theory could experimentally. The only current alternative working hypothesis is that of weak interactions we have ourselves repeatedly considered. As such, we strongly respect and accept it as true in some specific case but we prove here that the excluded volume theory is correct in its essence to the point to account for similar outcomes in cell and in crowder. We also delimitate the boundaries in relative size works which is an important result.

Reviewer #3 (Remarks to the Author) & answers

The manuscript "The many facets of the influence of macromolecular crowders on protein stability reveal that low temperature unfolding is intrinsically different from high temperature unfolding" by Alfano et al. reports on crowding studies on the folding stability of yeast frataxin. The innovative aspect of this study is that here crowding effects on both cold and heat denaturation are considered. The authors rationalize their experimental findings by analyzing the crowding effect on the temperature dependence of the folding free energy. In particular, they choose to quantify the free energy changes by a single parameter, which they define as the area enclosed by the individual curves with respect to dilute solution. They show that this is a much more powerful approach compared to numerous previous studies (summarized by their recent review in Curr. Opin. Struct. Biol.) that considered the effects of crowding agents on the melting temperature only, which could be misleading.

Thus, this is a very insightful study that will be interesting to a broad community of scientist that study in general the effect of cosolutes on protein folding. Thus, I highly recommend its publication but have three suggestions that should be addressed before publication.

We thank the reviewer for the kind words of appreciation and answered specific points as follows:

1.) The authors have great expertise in analyzing the folding of yeast frataxin from their previous work and the data appear to be robust and convincing. However, in the manuscript the authors do not present any error discussion e.g. of the thermodynamic quantities in Table 1. This would be very useful to compare to other approaches to determine (and extrapolate) $dG(T)$ e.g. by NMR or calorimetry (which the authors suggest in the discussion to be used for analysis of proteins other than yeast frataxin). Especially for the high concentrations of crowders in Figure 1 it seems difficult to me to determine T_c .

The reviewer is right. We have introduced an explicit treatment of errors.

2.) The authors exclude from their NMR measurements non-specific interactions between the crowder and the protein. However, they measure significant changes in enthalpy (Table 1) upon increase of crowder concentration. Please discuss this.

We thank the reviewer for raising this problem, we should have discussed it in the first place because it is very important.

The core of the problem is that it is difficult to conceive a change in stability (i.e. in free energy) of the protein that does not imply a change in enthalpy, whatever the origin of the change in stability. In other words, even if the increase in stability is only due to volume exclusion it will indirectly affect the difference in enthalpy between the folded and the unfolded species. It is a common but wrong belief that volume exclusion can cause a purely entropic change. If a compact species (the folded one) is favoured it is inevitable to favour all intramolecular interactions. It is precisely because of this that we tried to disentangle direct interactions (i.e. enthalpy changes due to a direct interaction between the crowder and the protein) from indirect effects by recording NMR spectra of Yfh1 in the presence of a crowder concentration (5%) sufficient to induce massive chemical shift changes if weak interactions are present, yet not high enough to cause significant volume exclusion.

We have introduced a discussion on the changes in enthalpy.

**We have previously addressed this problem in Sanfelice et al. (2013):
See also appendix to this paper for a formal statistical thermodynamics justification.**

3.) It may be beneficial to compare their approach to quantify the crowder induced changes of $dG(T)$ by the integral I/I_0 to a recently published similar approach by Senske et al., Phys. Chem. Chem. Phys., 18, 29698-29708, 2016. In their work the authors used 2 parameters to quantify the respective shifts of the curves for different cosolutes including different salts. This allows to distinguish a shift to the "right " as for e.g. sorbitol from a shift to the left e.g. propanol which is not possible for the integral approach. Maybe the authors can extend their approach to such cases.

We thank the reviewer for bringing to our attention this very interesting paper. It is difficult to insert a discussion such as that described in the paper by Senske et al. in the present manuscript. We shall consider it in future work.

REVIEWERS' COMMENTS:

Reviewer #1 (Remarks to the Author):

The revised manuscript adequately addresses the issues raised.

Reviewer #3 (Remarks to the Author):

My concerns and comments were satisfactorily addressed and I recommend the paper for publication.

REVIEWERS' COMMENTS:

Reviewer #1 (Remarks to the Author):

The revised manuscript adequately addresses the issues raised.

Reviewer #3 (Remarks to the Author):

My concerns and comments were satisfactorily addressed and I recommend the paper for publication.

We thank both referees for their positive comments